# Zero-Shot Controllable Image-to-Video Animation via Motion Decomposition

## ABSTRACT

In this paper, we introduce a new challenging task called Zero-shot Controllable Image-to-Video Animation, where the goal is to animate an image based on motion trajectories defined by the user, without fine-tuning the base model. Primary challenges include maintaining consistency of background, consistency of object in motion, faithfulness to the user-defined trajectory, and quality of motion animation. We also introduce a novel approach for this task, leveraging diffusion models called Img2VidAnim-Zero ($IVA^0$). $IVA^0$ tackles our controllable Image-to-Video (I2V) task by decomposing it into two subtasks: 'out-of-place' and 'in-place' motion animation. Due to this decomposition, $IVA^0$ can leverage existing work on layout-conditioned image generation for out-of-place motion generation, and existing text-conditioned video generation methods for in-place motion animation, thus facilitating zero-shot generation. Our model also addresses key challenges for controllable animation, such as Layout Conditioning via Spatio-Temporal Masking to incorporate user guidance and Motion Afterimage Suppression (MAS) scheme to reduce object ghosting during out-of-place animation. Finally, we design a novel controllable I2V benchmark featuring diverse local- and global-level metrics. Results show $IVA^0$ as a new state-of-the-art, establishing a new standard for the zero-shot controllable I2V task. Our method highlights the simplicity and effectiveness of task decomposition and modularization for this novel task for future studies.

## CCS CONCEPTS

• **Computing methodologies** → *Computer vision tasks.*

## KEYWORDS

Image-to-Video Animation, Controllable Video Generation

## 1 INTRODUCTION

The rising demand for controllable video generation underscores the desire of users to create videos for an increasing list of applications, such as personalized advertisement, educational content generation, visualization of imagination through user-generated content in social media, and entertainment content such as a short movie, where precise control of motion may be desired. While recent developments have made commendable strides in video generation from text prompts [14, 16, 18, 55, 61, 85], most works

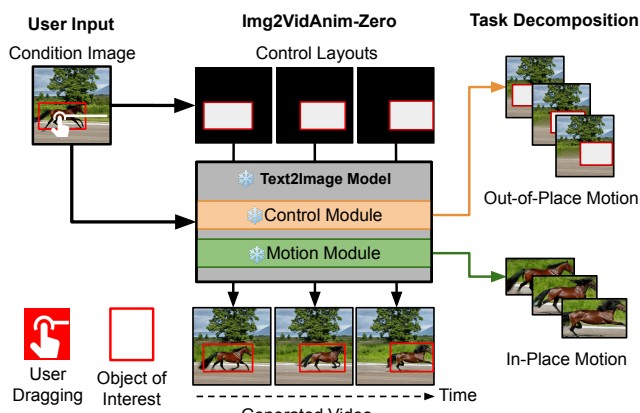

**Figure 1: Img2VidAnim-Zero for zero-shot image-to-video animation based on motion trajectories from the user. We decompose controllable Image-to-Video generation as two subtasks: (1) out-of-place motion generation, and (2) in-place motion animation which can be solved by leveraging existing modules pre-trained on other task-specific data.**

do not allow users to control the finer-grained details easily and interactively (*e.g.*by drawing trajectories or defining layouts).

Current developments in controllable video generation focus on the Image-to-Video generation task (I2V) [4, 21, 35, 63, 70, 74, 83]. I2V starts from a given condition image, eliminating the ambiguity often encountered with Text-to-Video (T2V) generation, enabling more diverse video animation based on additional conditions (*e.g.*text [70], trajectory [4, 74], or reference video [83]). As a result, I2V blends precision, versatility, and a more user-friendly set-up, positioning itself as a promising direction for controllable video generation. Recent I2V methods have utilized models trained on massive data with pre-extracted motion features [35, 63, 70, 74, 83]. However, ensuing challenges arise on: i) Computational resources: Even with efficient schemes like parameter-efficient fine-tuning [20], training models to understand new control conditions (e.g., motion vectors, trajectories) remain resource intensive. ii) Data collection: Acquiring data with meticulously annotated conditions can be expensive. Given these escalating costs, a pressing question is: *Can we devise a more cost-effective but still controllable I2V model?*

In this paper, we present Img2VidAnim-Zero ($IVA^0$) for Zero-shot Controllable Image-to-Video Animation without any I2V training data. The input set consists of the condition image and motion trajectories for objects of interest, represented by sequences of bounding box layouts. Our approach, as illustrated in Fig. 1 is to simplify the controllable I2V task by decomposing it into two atomic tasks. (1) **Out-of-place Motion Generation** focuses on determining the coarse layout of objects' obvious displacement throughout the video frames and (2) **In-place Motion Animation**

ensures consistency while facilitating plausible, smooth motion (pixel-level changes) for user-dragged in-box objects across the frames. We have identified that each atomic task can be tackled with pre-existing modules from the Latent Diffusion [50] family, leading to our goal of *Zero-shot* Controllable I2V Animation.

As shown in Fig. 1, our $IVA^0$ is based on 3 core components: (1) a pre-trained text-to-image model [50], (2) the Control Module (CM), and (3) Motion Module (MM). *Out-of-place motion* generation is formulated as a layout-to-image generation task, achieved by inserting Gated Self-Attention Layers [34] as a layout control module, leveraging bounding box layouts for precise object placements. We refer to this as the Control Module (CM). *In-place motion* animation is achieved by adopting Temporal Attention Layers [16], from the text-to-video generation task. We refer to this as the Motion Module (MM). It maintains the consistency of the selected objects by applying self-attention across frames, leading to a realistic and smooth transition of objects from one frame to the next. Notably, CM and MM are pre-trained on corresponding task-aligned datasets without any I2V-specific training. We further propose an efficient *Motion Afterimage Suppression* (MAS) scheme that generates frames via alternating different inpainting operations to reduce *afterimage*[1] hallucination objects that could be left trailing behind the motion trajectory, while maintaining a reasonable background.

Lastly, the proposed novel task requires a corresponding benchmark dataset for evaluation. While prior work [21] evaluates controllable I2V on synthetic datasets [13, 30] with limited quantitative metrics, we construct a new test bed with diverse objects, annotated control layouts, and more concrete metrics. We assess controllable I2V across local object aspects (including control accuracy, appearance and motion consistency, and object residual), global scene aspects (including scene consistency, and video quality), and human evaluation. We compare our $IVA^0$ with other strong I2V models [63, 70] that are end-to-end trained with massive data on the I2V task. This comprehensive evaluation reveals our zero-shot $IVA^0$ to be superior across 9 metrics, setting a new state-of-the-art on the proposed I2V benchmark. In summary, our main contributions are:

- Novel task of Zero-shot I2V Animation based on user-defined layout trajectories, along with novel approach IMG2VIDANIM-ZERO ($IVA^0$)
- Novel controllable I2V benchmark, enriched with diverse visual content and annotated control layouts (will be released), evaluated models across diverse dimensions.
- The IMG2VIDANIM-ZERO achieves competing results on the zero-shot I2V task. Our quantitative and qualitative results highlight the effectiveness and potential of our modular task decomposition idea for future controllable I2V studies.

## 2 RELATED WORKS

**Video Generation.** As a generative task with promising prospects, video generation has been a popular research topic. Early efforts [51, 58, 60, 77] focus on unconditional generation that is based on the vector initialized from a pre-defined probability space (*e.g.*Gaussian distribution). Recent works introduce various generation conditions and can be roughly categorized into: i) Text-to-Video generation (**T2V**) [1, 14, 16, 18, 23, 28, 36, 37, 55, 64, 85]: where descriptive text

is used as input to guide the generation process, ii) Video-to-Video generation (**V2V**) [8, 22, 40, 47, 67, 68, 73]: wherein a reference video informs the structure of the generated video, and iii) Image-to-Video generation (**I2V**) [4, 21, 27, 35, 44, 63, 70, 74, 79, 82, 86]: which uses a single or a series of images as the basis to produce a continuous frame sequence. We focus on the I2V formulation, which provides a clear visual starting point compared with T2V and gives more flexibility compared with V2V. Our work aims to inject layout-based controllability into the I2V.

**Controllability in I2V Generation.** Controllable video generation has become increasingly popular. Many efforts centered around encoding images and motion trajectories, mainly for human movement [2, 4, 5, 15, 69], editing a reference video through fine-grained control (e.g. dragging, depth/edge/pose maps) [12, 41, 54, 57]. Recent advancements include [74], which allows fine-grained object motion through user-defined trajectories, leveraging extensive video data, extra motion feature extraction, and multi-scale fusion modules. [63] is adept at synthesizing videos based on various combinations of appearance and motion patterns, given its training with varied spatio-temporal conditions. [35] introduces a neural stochastic motion texture for still images, ideal for objects with limited motion. Latest developments like [24, 36, 37] incorporate LLM layout planning into generation (and the former also introduces consistency in long-video generation) but these works focus on T2V generation. Very recent studies [9, 11, 25, 49, 62, 65, 66, 68, 72] make great progress in fine-tuning the model to be aware of diverse control conditions (e.g. detailed textual prompts, trajectories, boxes, and reference video) to animate an image. Furthermore, some other recent works adopt the efficient zero-shot setting, but they are suffering from controllability [75], focusing on T2V with extra LLM planning [56], or conducting extra DDIM operation [6]. Our method is not text-based or fine-tuned for the task, but an image-based zero-shot generation without any I2V-specific training or DDIM inversion. In addition, our method can also combine with these LLM planners to generate layout trajectory/sequence based on text for a more diverse control condition input.

## 3 METHOD

In this section, we introduce our $IVA^0$ model in detail. We first discuss the formulation of inpainting based on latent diffusion [50], which serves as the foundation of our model (Sec. 3.1). We then present how we build up $IVA^0$ by decomposing controllable I2V into sub-tasks, which can be addressed with out-of-place and in-place motion modules in the Latent Diffusion family (Sec. 3.2).[2] Finally, we elaborate on our Motion Afterimage Suppression (MAS) schema to eliminate object afterimage hallucination (Sec. 3.3).

### 3.1 Preliminary: Latent Diffusion for Inpainting

Our objective is to animate a static object, facilitating its transition from an initial to the subsequent position based on any user-defined layout trajectory. We frame it as an inpainting task for controlling such object movement, which involves: (1) replacing the original object location with the background, while (2) inpainting the object based on layout. To accomplish this, our $IVA^0$ is constructed on the

---

[1]https://en.wikipedia.org/wiki/Afterimage

[2]In our implementation, we adopt Stable Diffusion which is an improved variance of the Latent Diffusion Model: https://github.com/Stability-AI/StableDiffusion.

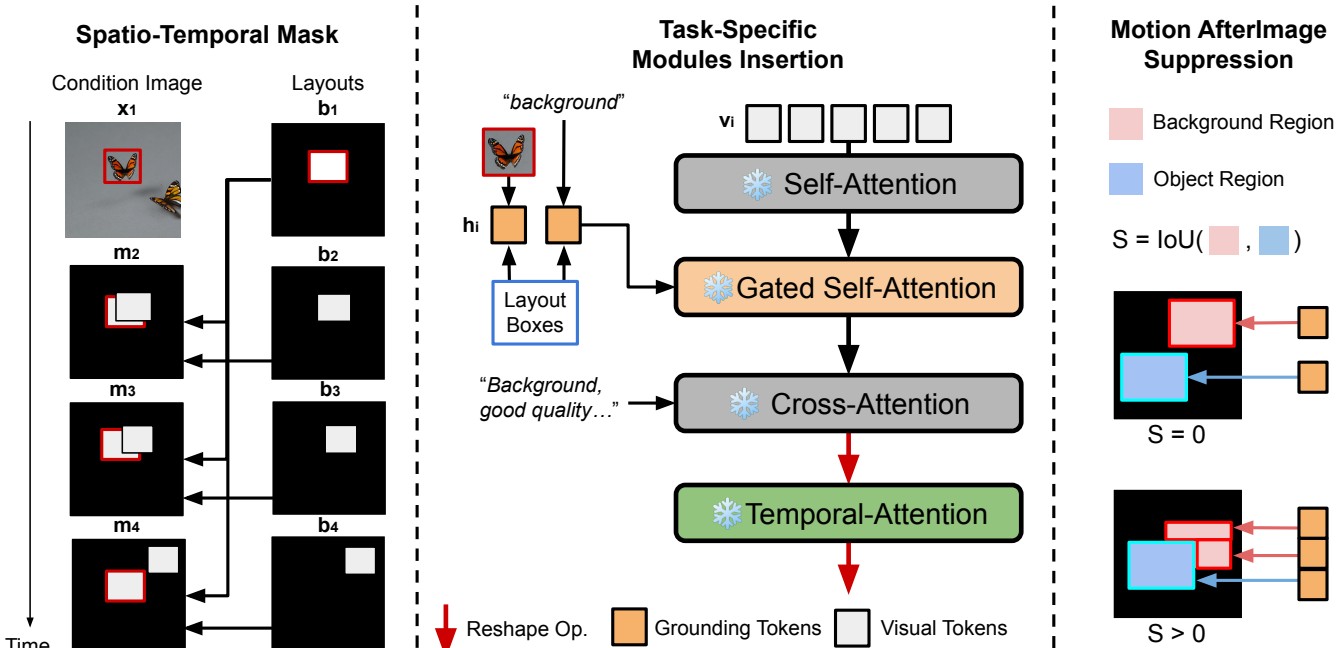

**Figure 2: Left: Our spatio-temporal inpainting masks cover the starting position of objects and their target locations in each frame. Middle: To handle various atomic tasks in controllable I2V, we integrate different task-specific modules. We apply gated self-attention layers as the control module for generating out-of-place motion, while using temporal-attention layers as the motion module for in-place motion animation. Right: We introduce Motion Afterimage Suppression (MAS), which uses object size and IoU to decide whether to inpaint the background with additional grounding tokens. This approach aims for enhanced inpainting quality with reduced afterimage hallucination.**

inpainting version of Latent Diffusion [50], a publicly available pre-trained text-to-image model. The Latent Diffusion model comprises three key components: (1) Autoencoder that maps the image from pixel space to latent embedding, based on which the Diffusion module operates; then projects the embedding after denoising steps back to pixel space; (2) Text encoder that encodes a prompt into embedding for Text-to-Image conditioning; (3) U-Net for noise diffusion, which iteratively conducts denoising in the latent space, guided by timestamps and prompt embedding.

The inpainting task leverages the Latent Diffusion model to modify a masked image region based on the given textual conditions. This mask is represented as a 1-channel binary mask as additional input together with the condition image. The latter delivers essential context for the un-inpainted sections and is derived by processing a condition image $x_{con}$ through the encoder. To adapt to these extra inpainting conditions, the diffusion U-Net incorporates five extra channels in its initial convolution layer. Given a condition image $x_{con}$, a text prompt $p$, and a binary mask $m$, the inpainting model generates an image. In the following sections, as depicted in Fig. 2, we delineate the integration of control conditions and how we handle basic atomic tasks with this text-to-image inpainting model to achieve controllable I2V.

## 3.2 Zero-Shot Layout-Conditioned I2V

In IVA$^0$, we introduce a controllable Image-to-Video generation model that leverages user-provided spatio-temporal object layouts.

Given an initial frame $x_1$ as condition image $x_{con}$, users can animate specific objects by providing a trajectory of the object. This trajectory, in our method, is represented as a sequence of bounding boxes: $(b_1 \cdots b_t)$, where $t$ refers to the number of frames. Each box $b_i$ is a 4-dimensional vector indicating the top-left and bottom-right coordinates of the box. For simplicity, we focus on animating only a single object at a time. But, IVA$^0$ is versatile and can be extended to multiple objects simultaneously when provided with corresponding layouts. The detailed model pipeline is elaborated as follows:

**Layout Condition via Spatio-temporal Masking:** A future frame $x_i$ is generated based on the initial frame $x_1$ ($x_{con}$) and layout boxes via the T2I model. When transitioning the object from its position $b_1$ to $b_i$, we expect the object to move to the desired location with smoothly interpolated motion and consistent appearance for both the foreground object and background context. This necessitates inpainting $x_i$ in two regions: (1) eliminate the object from the original region at $b_1$ as part of the background, and (2) add the object to the new region $b_i$. So we create an inpainting mask for each frame by simultaneously masking out both starting region $b_1$ and target region $b_i$. As illustrated in the left of Fig. 2, from the spatio-temporal layout sequence $(b_1 \ldots b_t)$, we construct the spatio-temporal masking sequence $\mathbf{M} = (m_1 \ldots m_t)$, with each $m_i = b_1 \cup b_i$.

**Out-of-Place Motion Generation:** An important task of our model is to generate out-of-place motion, given the spatio-temporal

masks **M**. This can be formulated as a layout-conditioned generation similar to [10, 34, 81, 84], which requires generating an image following a layout condition. Thus, we adopt the design of gated self-attention proposed in GLIGEN [34], a layout-to-image generation model. GLIGEN encodes object box coordinates into special *grounding tokens* and fuses grounding information with visual tokens via extra gated self-attention that is added before each cross-attention layer in the text-to-image model. Specifically, as shown in the middle of Fig. 2, we insert Gated Self-Attention layers inherited from [34] with copied weights, as our Control Module. The control module then utilizes grounding tokens that encapsulate both the appearance of the object and its box coordinates, enabling precise placement of the object in the desired location. We streamline the process by using the same CLIP model [48] as an image encoder to extract regional image features of the cropped object. The box coordinates $b_i$ are projected into continuous embedding with Fourier transform function as in [42], controlling the spatial location. Thus, for the frame at time $i$, layout tokens $h_i$ are derived by integrating these conditions via a linear projection layer. These tokens then interact with the visual tokens of the same frame using gated self-attention, ensuring accurate and contextually relevant out-of-place motion generation, such that[3]:

$$h_i = MLP(CLIP_{img}(crop(x_1, b_1)), Fourier(b_i)) \quad (1)$$

$$v_i = SelfAtt(concat(h_i, v_i)) \quad (2)$$

**In-Place Motion Animation:** Based on our observation, relying solely on the out-of-place inpainting strategy only produces a rudimentary "copy-paste" animation for objects (see Fig. 6), causing noticeable inconsistencies in their motions across frames. In order to pursue a smoother and authentic object-moving motion and ensure sustained visual coherence, we adopt an in-place motion animation module. Previous works [14, 28, 61, 63] show different inter-frame attention mechanism that helps this goal, but unanimously require large-scale pre-training from video data. We resort to a pre-trained motion engine [14], as illustrated in the middle of Fig. 2, and incorporate its Temporal Attention layers [14] with weights copied from the original Text-to-Video generation task, but for our controllable I2V task. This motion module enables better temporal consistency for both object appearance and motion via self-attention across frames. Specifically, given the sequential frame visual features $\mathbf{V} = (v_1 \dots v_t)$, where $\mathbf{V} \in R^{(t,h*w,c)}$, we reshape the feature axes and apply self-attention to the temporal dimension, where $w$, $h$, and $c$ refer to width, height, and feature channel.

$$\mathbf{V} = Reshape(SelfAtt(Reshape(\mathbf{V}))) \quad (3)$$

Both the control and motion modules are pre-trained on different task-specific data. They integrate capabilities from their original Layout-Conditioned Image Generation and Text-to-Video tasks. We incorporate these established foundations for our generation to avoid further re-training. Thus, our IVA[0] can controllably animate objects in an image without any Controllable I2V-specific fine-tuning. Sec. 4.2 contains more training details of these modules.

---

[3]Here the concatenation is for one attention block. Details in [34].

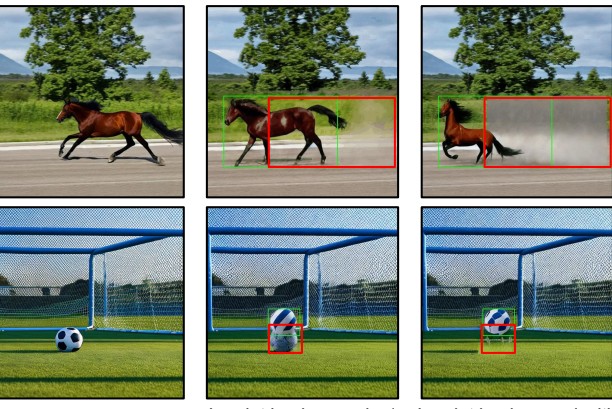

|  |  |  |
|---|---|---|
| Condition Image | Inpaint background w/o grounding token | Inpaint background with grounding token |

**Figure 3: Comparison of background inpainting with or without grounding tokens. The token-based method can eliminate the afterimage object hallucination issue (middle-bottom red box) but is weaker in handling larger regions (right-top red box) compared with the token-free one.**

## 3.3 Motion Afterimage Suppression

As our IVA[0] model is built upon the Text-to-Image inpainting model, we initially prompt the model with a fixed background-filling text prompt (shown in Fig. 2 middle) for background generation. However, as illustrated in Fig. 3 bottom-middle, we observed this approach sometimes results in an afterimage, a ghost-like residual hallucination after the object has moved (bottom row of Fig. 3). We further find out that this issue is linked to the motion module in our model. As evidenced in Fig. 6 (rows 2 and 3), these hallucinations occur when the motion module is used in the model. This is because most current motion modules [7, 16, 70] only generate limited-range, in-place motion. For this reason, when an object moves significantly from its original $b_1$, the temporal attention fails to maintain appearance consistency. Inversely, the temporal attention often wrongly produces an afterimage at $b_1$ in $x_t$.

To suppress such a motion afterimage, we experimented with using extra grounding tokens for the background generation. As illustrated in Fig. 2 middle, we set grounding tokens that encode both the bounding box and the word 'background' (via CLIP text embedding). In this case, we force the control module to generate a background at $b_1$ in frame $x_t$. This method successfully avoids afterimage hallucinations for small objects but struggles with large areas (see Fig. 3 top-right). This limitation likely stems from the control module's training on background reconstruction, which cannot handle large-region in-painting with a single token. To overcome this, we integrated two background generation approaches based on object size and Intersection over Union (IoU). Objects are first categorized by size (small, medium, large) with pre-defined area thresholds. Small objects use extra grounding tokens for background in-painting, whereas large objects do not. For medium-sized objects, as shown in Fig. 2 (right), we first calculate the IoU $S$ between $b_1$ and $b_t$, if $S > 0$, indicating overlap, the non-overlapping background areas are first divided into grids iteratively. The model

 

then in-paints each grid with background class tokens. Experiments show that our proposed MAS resolves the afterimage object hallucination while maintaining high-quality background generation.

## 4 EXPERIMENTS

Our experimental setup is detailed in this section, including proposed metrics & benchmark for quantitative evaluation (Sec. 4.1), implementation details (Sec. 4.2), results analysis (Sec. 4.3), and limitation discussion (Sec. 4.4). Details of benchmark data collection, human evaluation, and baseline implementation are in Appendix.

### 4.1 Evaluation

To comprehensively assess the performance of our model under controlled experimental conditions, we evaluate its effectiveness across various metrics. This evaluation encompasses both the local level (concentrating on objects achieving smooth motion, maintaining a consistent appearance, and following layout conditions), and the global level (focusing on generated video scenes matching the given condition image).

- **vIoU@R:** This evaluates the correctness of layout conditioning. We adopt this metric from action detection [71, 80], which calculates the spatio-temporal overlap between ground truth and GroundingDINO [39] detected boxes: i.e., if vIoU exceeds threshold $R$, then we consider the prediction to be a match, namely vIoU@R. In our experiment, we report results with $R = 0.3, 0.5$.
- **Smoothness:** We propose the "Smoothness" metric, which evaluates the smoothness of object transformations across video frames. We compare object similarity in consecutive frames using GroundingDINO for object detection and CLIP ViT-B/32 for image embeddings. Smoothness is calculated by averaging the similarity of these embeddings across all frames, with a higher score indicating smoother changes in appearance in the video.
- **Hallucination:** We propose the "Hallucination" metric, capturing the wrong afterimage generations. We first detect the target object class in each frame with GroundingDiNO. Then, we calculate the difference in object count between the generated and the condition image and sum normalized results over multiple frames for a video-level metric. This metric reflects the unexpected generation (e.g., extra object) or removal of an object.
- **SSIM & LPIPS:** These measure the structural similarity of generated frames with the condition image. As we are only interested in animating the object in the image while maintaining other regions, the higher structural similarity between the condition image and the generated frames means that the model can keep the background scene or non-interested regions unchanged. We adopt both non-parametric SSIM [19] and parametric LPIPS [78] to represent structural similarity.
- **FID & FVD:** These standard reference-based metrics ([17], [59]) quantify the visual appeal by comparing the sets of ground truth and generated videos. To compute FVD, we repeat and stack initial frames as pseudo-video for distribution gap computation.
- **Human Evaluation:** Automated metrics are not perfect. Hence, we include human evaluation as well. Annotators are asked to conduct a majority voting for the best-quality video considering appearance consistency of object/background, motion faithfulness and motion quality.

**Controllable I2V Benchmark:** Since our controllable I2V task focuses on the animation of objects, we evaluate the model on a testbed that contains ground-truth videos without camera motion, involves diverse objects, permits reasonable motion ranges, and includes controlled layouts. For this:

- We collect 100 images as initial frames, a mix of generated ones using Stable Diffusion [50], and real images from a public dataset [46]. These images feature diverse objects for animation.
- For each image, we manually annotate the start and end boxes for an object and interpolate them with intermediate boxes using a non-linear function as a trajectory.
- We also annotate each sample with a textual prompt, describing the desired motion of the object. In total, our testing set comprises 200 object control layouts & captions, paired with 100 images.

**Baseline Models:** We compare our method against two competing image-to-video generation baselines: 1) *VideoComposer* [63], a compositional video synthesis model that offers motion controllability conditioned on motion vector. It is pre-trained on *WebVid10M* [3] and *LAION-400M* [52]. It is based on the Video Latent Diffusion Model (VLDM) [61] that incorporates both 3D convolution and temporal attention. In the evaluation, we adapt our layouts into motion vector format for compatibility. 2) *DynamiCrafter* [70], a recent emerging image animation model that has been pre-trained on *WebVid10M* dataset [3]. It is also built on the VLDM. Since *DynamiCrafter* is unable to be conditioned by trajectory, we circumvent this by providing an extremely detailed prompt as the condition.

### 4.2 Implementation Details

**Model Implementation**: We choose Stable Diffusion-v1.4 as our base model, which is pre-trained on *LAION-400M* [52]. Our control module is derived from GLIGEN [34], pre-trained on various grounding datasets, including *COCO2014D, COCO2014CD, and COCO2014G.* Our motion module, derived from AnimateDiff [14], is pre-trained on *WebVid10M* [3]. The guidance scale was set to 7.5 and 25 steps were used for denoising with DDIM noise scheduler. 16-frame videos were generated for both our method and baseline methods. The resolution was set at 512×512 for generated videos. 5 random seeds were used for each layout trajectory; We categorize objects based on their area relative to the image: small objects occupy no more than 1/16, medium ones range between 1/16 and 1/5, and large ones exceed 1/5 of the image area. We use 5 random seeds (1, 2, 3, 42, 126) to get five unique generations for each image-box pair. Our noise scheduler adopts 0.00075 beta start, 0.012 beta end, and "scaled_linear" beta schedule. The negative prompt used for the generation was: "worst quality, deformed, extra object, extra human, distorted, disfigured, bad anatomy, disconnected limbs, wrong body proportions, low quality, illustration, oversaturated, cartoons, blurry, cropped, text."

**Baseline Implementation**: For DynamiCrafter [70], which does not support conditioning with layout trajectories, we prompt the model with the detailed motion-related text (see Appendix). For VideoComposer [63], which can use hand-crafted motion vectors as extra conditions, we first generate motion vectors from layout sequences and then use the vectors to generate video.

**Data Collection**: We collect both 20 real-world images from public tracking/segmentation datasets [46] and 80 generated images by

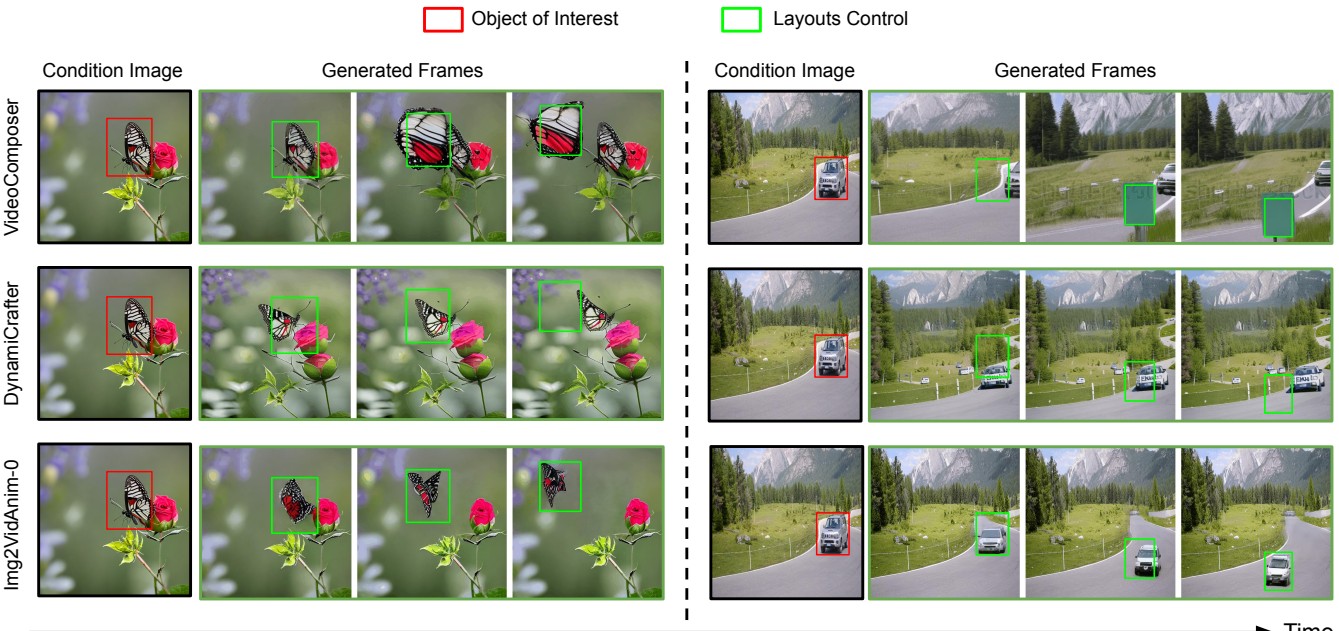

Figure 4: Qualitative comparison of our method and baseline approaches. The sample on the left is based on initialization with a generated image, while the sample on the right was initialized with a real image. Best viewed in color and zoomed in for more details.

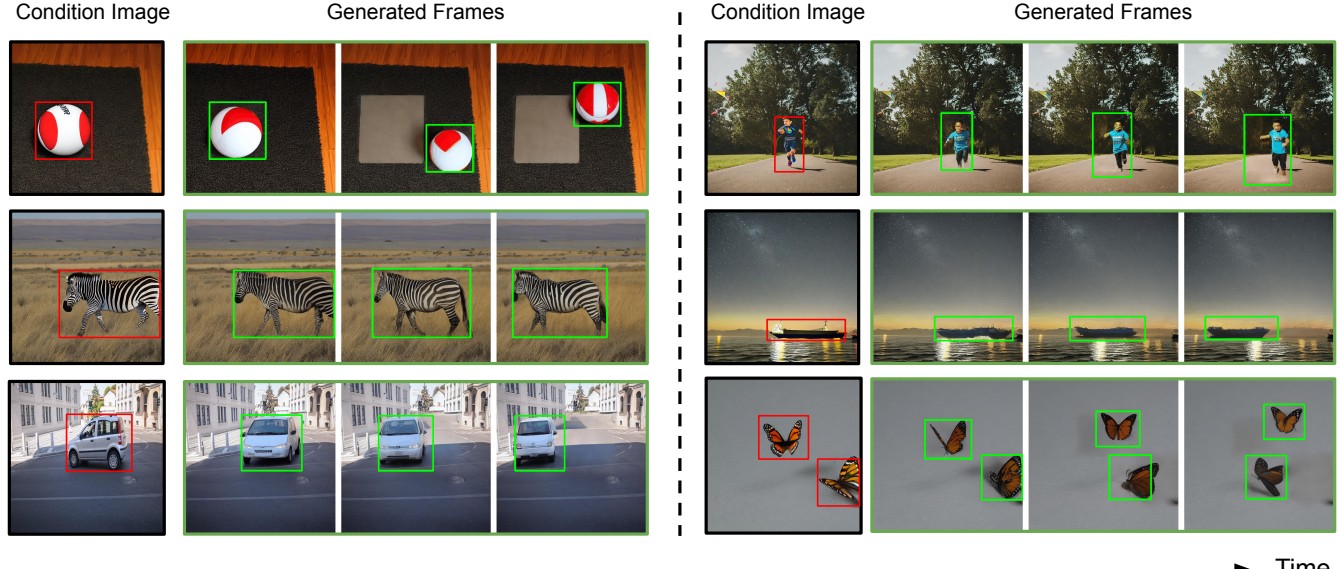

Figure 5: Sample results of the proposed IVA$^0$. The green boxes in the generated frames represent input control layouts. Best viewed in color and zoomed in for the best visualization. We provide more qualitative results in the Appendix.

StableDiffusion-2.0 [50]. We manually write text prompts for the StableDiffusion model to generate images. We collect images containing single/multiple objects and with an empty area that allows for obvious out-of-place object motion.

**Data Annotation**: To reduce the labeling burden, we only manually annotate key layout boxes for each image, and interpolate those key boxes to imitate user dragging and obtain layout sequence during inference. We also write detailed motion-related captions

**Table 1: Quantitative comparison with evaluated baselines on the proposed controllable I2V benchmark. Smooth.: object smoothness across frames. Hallu.: object hallucination in generated videos.**

| Methods | 0-Shot | Local Object | | | | Global Scene | | | | HumanEval |
|---|---|---|---|---|---|---|---|---|---|---|
| | | vIoU@0.3 ↑ | vIoU@0.5 ↑ | Smooth. ↑ | Hallu. ↓ | SSIM ↑ | LPIPS ↓ | FID ↓ | FVD ↓ | Win Rate↑ |
| VideoComposer [63] | N | 50.0 | 25.3 | 85.8 | 42.4 | 36.4 | 45.2 | 151.4 | 1790 | 31% |
| DynamiCrafter [70] | N | 21.4 | 5.4 | 89.2 | 42.5 | 25.9 | 56.0 | **132.1** | 1460 | 27% |
| $IVA^0$ (ours) | Y | **88.7** | **78.1** | **90.2** | **13.6** | **61.9** | **28.0** | 132.7 | **1352** | **42%** |

for each image-layout pair to prompt baseline models [70] that cannot take layout input. We provide data visualization (image, key boxes, caption) in Appendix.

**Grounding Model Implementation**: We use GroundingDINO-B [39] with Swin-B backbone and pre-trained on COCO [38], O365 [53], GoldG [26], Cap4M [32], OpenImage [29], ODinW-35 [31], RefCOCO [76] for grounding detection in the evaluation pipeline. The box threshold was set to 0.35, and the text threshold was 0.25.

**Human Evaluation**: As discussed in Sec. 4.1, we designed a pipeline for human evaluation. Specifically, we first generate 100 example sets, where each example set contains 1 condition image and 3 videos generated by different methods (our $IVA^0$ and 2 baseline methods, in a shuffled order). Then, we ask 5 raters to rank among these 3 generated videos from the following aspects: 1) Controllability: Which method follows the object layout the best (We only consider whether the object is present in the given layout)? 2) Background Consistency: which method best maintains the background shown in the condition image? 3) Motion faithfulness: which method demonstrates the most plausible object out-of-place motion & object consistency? For each question, the raters are asked to select the best video. Our goal is to calculate the win rates of models along these 3 questions.

## 4.3 Results

**Qualitative Results**: Fig. 4 shows two examples of generation with our method compared with baselines. In practice, we find it challenging to leverage *DynamiCrafter* to animate the object strictly following user instructions by purely prompting; e.g., the Lepidoptera and car can barely follow the box condition. Though *VideoComposer* shows good layout conditioning, it suffers from the hallucination of new objects or sometimes the removal of the target object; e.g., an extra Lepidoptera was not successfully eliminated in the original region. We find our $IVA^0$ follows the layout condition better than the baselines and also generates smooth motion with a more consistent object appearance. We also provide more samples generated by our $IVA^0$ in Fig. 5 with various animation conditions, including single/multiple objects, small/large objects, real/generated images, and simple/complex motion trajectories. We observe consistent conclusions across all combinations. However, we do observe all methods performing less satisfactorily for maintaining consistent object appearance across frames; e.g., in Fig. 5, appearance of children and vehicles is altered in the generated frames. We conjecture for these reasons: 1. It still remains challenging for the T2I model to produce customized & consistent objects without any weight optimization; 2. CLIP embedding does not capture information about the object comprehensively.

**Quantitative Results Analysis**: Tab. 1 contains quantitative results. Our $IVA^0$ shows leading results on local-object metrics, i.e., vIoU, Smoothness, and Hallucination. Specifically, $IVA^0$ largely surpasses *VideoComposer* and *DynamiCrafter*: 88.7 vs. 50.0 and 21.4 when $R$=0.3. This verifies that our $IVA^0$ possesses the capability of precise layout control. In addition, we notice that $IVA^0$ has a higher Smoothness score: 90.2 vs. 89.2 by *DynamiCrafter*, indicating that our $IVA^0$ produces objects with smoother changes across frames. Noticeably, the obvious lead in the Hallucination metric score (13.6 vs. 42.5) also consolidates this conclusion. When evaluated at the scene level, we observe very aligned results. $IVA^0$ leads the baselines on SSIM, LPIPS, and FVD, while showing a close gap with *DynamiCrafter* on FID: 132.7 vs. 132.1. We note that both FID and FVD compute the distribution gap. Since our evaluation is just based on a single initial frame, the slight gap here just indicates that the distribution of the produced video frames is not evidently deviating from the initial frame. Furthermore, our human evaluation study shows the superiority of the animation produced by our method: 42% win rate versus 31% and 27% for both baselines. Details on human studies can be found in the Section 4.2.

**Ablation Study.** We now assess the effectiveness of each module and the Motion Afterimage Suppression strategy. As illustrated in Fig. 6, we observe that CM largely improves the control ability of $IVA^0$ on vIoU@0.3: 26.7 vs. 88.7. Adopting a motion module explicitly improves motion smoothness, which is reflected in the Smoothness and FVD metrics. We observe that the motion module has negative impacts on FID, LPIPS metrics, which aligns with our expectations as it is interpolating frames with diverse motion, inevitably diverging it from the initial frame used as ground truth on these metrics. The MAS module largely improves the baseline performances on a series of local object metrics: 88.7 vs. 87.3 on vIoU@0.3 and 13.6 vs. 21.2 on the Hallucination metric. It should be noted that while integrating the motion module improves temporal video metrics like Smoothness and FVD, it also diminishes the grounding capability (measured by VIoU) facilitated by the control module. It is a trade-off for such a zero-shot model without extra alignment for injected modules. The full model achieves a synergistic balance between these capabilities across modules, leading to better overall quality. Fig. 6 further showcases more examples of generated images with each module.

## 4.4 Limitations & Discussion

Despite of the generated video, our zero-shot model $IVA^0$ still faces several challenges, as shown in Fig. 7, as following:

**(1) Inconsistent & Missing Object**: $IVA^0$ struggles to maintain appearance consistency of object appearance w.r.t. target object

Table 2: Ablation study on our method. CM: Control Module. MM: Motion Module. MAS: Motion Afterimage Suppression.

| MM | CM | MAS | Local Object | | | | Global Scene | | | |
|---|---|---|---|---|---|---|---|---|---|---|
| | | | vIoU@0.3 ↑ | vIoU@0.5 ↑ | Smooth. ↑ | Hallu. ↓ | SSIM ↑ | LPIPS ↓ | FID ↓ | FVD ↓ |
| - | ✓ | ✓ | **90.2** | **85.7** | 88.4 | 14.4 | **63.2** | **25.9** | **118.4** | 1440 |
| ✓ | - | ✓ | 26.7 | 21.6 | 70.1 | 43.5 | 58.4 | 32.8 | 182.0 | 1388 |
| ✓ | ✓ | - | 87.3 | 77.5 | 89.5 | 21.2 | 62.4 | 27.4 | 133.6 | 1369 |
| ✓ | ✓ | ✓ | 88.7 | 78.1 | **90.2** | **13.6** | 61.9 | 28.0 | 132.7 | **1352** |

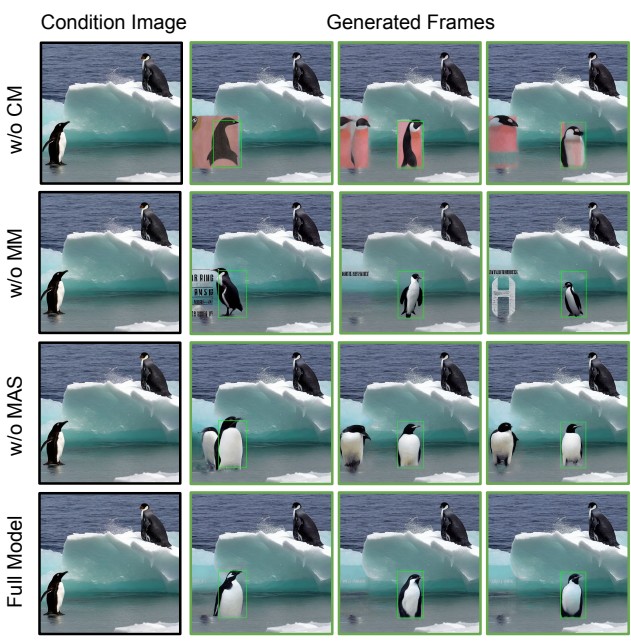

Figure 6: Qualitative images are produced by our IVA$^0$ with and without each module. The last row shows the images produced by the combination of all modules.

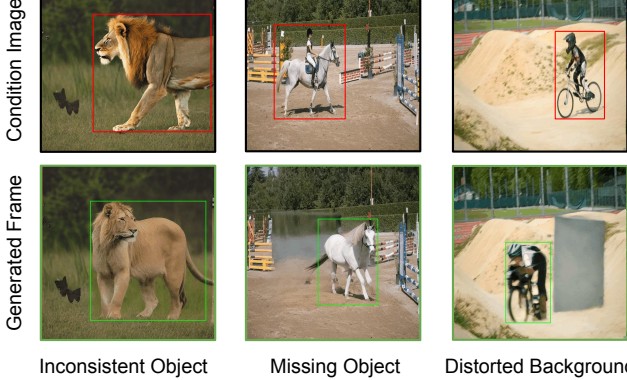

Figure 7: Examples of failure cases in our model include producing objects with an inconsistent appearance, missing objects, or distortions in the background.

in the initial frame, and it tends to overlook secondary objects when multiple objects are present in a single box (*e.g.* woman on the horse). This discrepancy arises possibly because CLIP image embedding adeptly captures mainly high-level features but neglects the finer low-level features (i.e., texture, color, and shape). Such difference is further amplified when we apply IVA$^0$ to the real-world images due to the potential domain gap present in the pre-training text-to-image model [43]. To quantify this, we computed the SSIM↑ between ground-truth objects and the reconstructed images. The results indicate that inpainted objects share only 22% SSIM↑ score with the GT, highlighting an unavoidable loss of low-level details. **(2) Distorted background**: it inaccurately generates backgrounds (*e.g.* gray patch behind the man on bike), especially for real images with complex scenes. We attribute those to a lack of extra training/alignment with the pre-trained motion and control modules for the base T2I model. It weakens the base model's inpainting and generation ability. This mismatch is further exacerbated when applied to real-world images due to a potential domain gap in pre-trained models [33, 43]. Besides, our model is currently limited to animating foreground objects, and cannot modify the background in case users want to animate the background too or incorporate camera motion.

However, we note that the spatio-temporal consistency across scenes and objects still remains an open challenge for all existing T2V/I2V models, which rely on large-scale pre-training as a possible solution [45]. Our efforts focus on building an efficient zero-shot T2V model without any tuning. As part of future directions, we suspect (1) integration with a text-to-video backbone that applies temporal modules, e.g., 3D convolution, (2) fine-tuning with I2V data will further enhance consistency, (3) integration with more control conditions, e.g., segmentation masks. We also attempt to mitigate those bad generations with recent popular one/few-shot tuning ideas in text-to-video generation work (e.g., [67, 68]) (see more details in Appendix) and give more insights. In all, addressing these challenges remains an open topic for future research.

## 5 CONCLUSION

In this paper, we introduced IMG2VIDANIM-ZERO (IVA$^0$), a Zero-shot Image-to-Video (I2V) method without task-specific I2V training. By harnessing existing text-to-image Diffusion modules and integrating Gated and Temporal Attention layers, IVA$^0$ facilitates accurate and seamless video generation from the specified motion trajectory based on bounding boxes. Our novel I2V benchmark underscores IVA$^0$'s leading performance, showcasing its potential for future video generation applications.

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
