# OpenReview forum: "Zero-Shot Controllable Image-to-Video Animation via Motion Decomposition"
_acmmm.org/ACMMM/2024/Conference — MM2024 Poster_

### Official Review · Reviewer_xRWh · 2024-05-19

**Rating:** 2
**Confidence:** 2

**Summary:**

In this paper, the authors present the new challenging task of zero-shot controllable image-to-image animation and a novel zero-shot approach Img2VidAnim-Zero (${IVA}^{0}$) to allow users to control the trajectory of the object's motion in an image. ${IVA}^{0}$simplifies the I2V task by decomposing it into two subtasks: out-of-place and in-place motion animation. 1): Out-of-place Motion Generation focuses on determining the coarse layout of objects' obvious displacement throughout the video frames; 2) In-place Motion Animation ensures video consistency while facilitating object smooth motion. Experiments demonstrate the effectiveness of ${IVA}^{0}$.

**Strengths:**

1. In this paper, the authors propose a user-defined layout trajectory based zero-shot method ${IVA}^{0}$.
2. In this paper, the authors present a novel controlled I2V benchmark that evaluates models across different dimensions.

**Limitations:**

1. Peekaboo [1] can also control the trajectory of objects in the image, but there is no Peekaboo in the comparison method.
2. How does it work if the trajectories of multiple objects intersect or overlap at the same time?
3. The article combines different modules from different baselines and is not fine-tuned to each other. Are there any conflicts between them? How were these conflicts resolved?
4. In lines 432-464 of the article, it is mentioned that the article used CLIP text embedding to generate the background. However, it is difficult for the text to represent the detailed content of the background, so please explain how to keep the background in the generated frames unchanged from the background in the original frames.
5. In the Human Evaluation, the raters were only 5 whether it was not generalizable.
6. The appearance of the object changes in the generated frames, and the article explains that this is due to the fact that CLIP image embedding is not capable of capturing low-level features such as shape. So have other embedding methods besides CLIP been explored?

[1] Jain Y, Nasery A, Vineet V, et al. PEEKABOO: Interactive Video Generation via Masked-Diffusion[J]. arXiv preprint arXiv:2312.07509, 2023.

**Suitability:**

3

---

### Official Review · Reviewer_sRZK · 2024-05-22

**Rating:** 3
**Confidence:** 3

**Summary:**

This paper introduces zero-shot controllable image-to-video animation, the goal is to animate an image based on motion trajectories defined by the user. The authors decompose this task as two sub-tasks: out-of-place motion generation, and in-place motion animation. Experiments verify the proposed model's effectiveness.

**Strengths:**

1. A new-task is proposed, namely zero-shot controllable image-to-video animation.
2. The authors propose to apply gated self-attention layers as the control module for generating out-of-place motion, and use temporal-attention layers as the motion module for in-place motion animation.
3. The writing of this paper is good.

**Limitations:**

1. Quantitative comparison is limited. The authors should compare with more related baselines, for example, [1][2][3].
2. The paper lacks the visualization of predicted motion vectors. Specifically, the visualization comparison between out-of-place motion and in-place motion.
3. The paper does not provide running speed of the proposed method.


[1] AnimateZero: Video Diffusion Models are Zero-Shot Image Animators. (2023).
[2]. MotionZero: Exploiting Motion Priors for Zero-shot Text-to-Video Generation.
[3] Motion-Zero: Zero-Shot Moving Object Control Framework for Diffusion-Based Video Generation.

**Suitability:**

2

---

### Official Review · Reviewer_weU7 · 2024-05-23

**Rating:** 4
**Confidence:** 3

**Summary:**

The article introduces a novel approach named Img2VidAnim-Zero ($IVA^0$) for zero-shot controllable image-to-video animation generation. This method aims to animate a still image through user-defined motion trajectories without the need for fine-tuning the underlying model. The paper addresses this challenge by decomposing the image-to-video (I2V) task into two sub-tasks. By integrating contributions from methods such as Stable Diffusion, GLIGEN, and AnimateDiff, the approach achieves a direct-use capability without additional training and introduces a MAS optimization to enhance the generative results.

**Strengths:**

1. This paper skilfully uses a combination of existing methods to accomplish a new task, and although it somewhat lacks technological innovation, it skilfully applies existing technology to accomplish a meaningful task.
2. The article designs a new controllable I2V benchmark that includes diverse objects and annotated control layouts, and proposes multiple local and global level metrics. Experimental results show that IVA0 achieves a new optimum on the zero-sample controllable I2V task and sets a new standard for future research.
3. The paper is well written and the visualisation illustrates the effect of the work well.

**Limitations:**

1. There is a certain curiosity regarding how the outcomes of the proposed method compare to those obtained by editing the final frame of a video through image editing techniques, followed by the application of frame interpolation technologies, such as those offered by DynamiCrafter for generating video between two frames. A more appropriate benchmark for evaluating DynamiCrafter might involve directly editing the input images using image editing method to depict the final post-movement position as the video's last frame. Subsequently, the original image would serve as the first frame, and DynamiCrafter would be employed for generative frame interpolation. It would be insightful to compare the results of such a method with those presented in this paper.
2. This paper lacks a subjective and objective evaluation of the consistency in the appearance of moving objects. It may be beneficial to supplement the assessment with metrics such as CLIP-I and DINO-I for comparison with other methods, thereby further evaluating the consistency of the objects' appearance.
3. Does MAS require different set parameters on different data sets? It will limit the ease of use of the method.

**Suitability:**

3

---

### Official Review · Reviewer_UBeH · 2024-05-24

**Rating:** 4
**Confidence:** 4

**Summary:**

The article introduces a new technology called Img2VidAnim-Zero (IVA0), which achieves zero-shot controlled animation conversion from images to videos. IVA0 decomposes the task into two subtasks: “out-of-place” and “in-place” motion animations. By leveraging existing layout-conditioned image generation and text-conditioned video generation methods, IVA0 allows users to create animations based on their defined motion trajectories without fine-tuning the underlying model

**Strengths:**

Introducing a Novel Controllable Image-to-Video (I2V) Benchmark

This paper proposes a novel task decomposition method by breaking down the animation task into subtasks for “out-of-place” and “in-place” motion animations. It effectively leverages pre-trained models to achieve zero-shot controlled animation generation.

IVA0 combines gated self-attention and temporal attention layers, enhancing the precision of object placement and the smoothness of motion.

**Limitations:**

There is no user study to further validate the reliability of the experimental results.

As shown in Figure 5, the preservation of the subject in the Condition Image is poor.

**Suitability:**

3

---

### Meta-Review · Area_Chair_Lb4Z · 2024-07-05

**Recommendation:** Accept (Poster)
**Confidence:** 5

**Metareview:**

After considering all reviews, the rebuttal, and the subsequent discussion, the consensus is to accept the paper. The authors are high suggested to address the concerns raised by the reviewers in the final version.